

# Organic fertilizer type and dose affect growth, morphological and physiological parameters, and mineral nutrition of watermelon seedlings

Qianwen Zhang, Joseph Masabni and Genhua Niu

Texas A&M AgriLife Research and Extension Center at Dallas, Texas A&M University, Dallas, TX, United States of America

## ABSTRACT

**Background**. Organic agriculture has grown rapidly in recent years due to its environmental friendliness, sustainability, and improved farm profitability. Transplants are commonly used for fruits and vegetables to achieve consistent quality, uniformity, and easy field spacing control. The efficacy and optimal amounts of fertilizers for organic transplant production need to be investigated.

**Methods**. The effects of three organic fertilizers (Sustane 4-6-4, Nature Safe 7-7-7, and Dramatic 2-4-1) and one conventional fertilizer Peters Professional 20-20-20 (Conventional) with four doses (nitrogen (N) content was matched among fertilizers in each level, as 0.14 g/L, 0.28 g/L, 0.56 g/L, and 0.84 g/L N, respectively) on watermelon seedlings were compared in this study.

**Results**. The results showed that all organic fertilizer treatments were not significantly different from the Conventional group in terms of watermelon germination. The only exception was the highest dose of Sustane 4-6-4 (0.84 g/L N) which decreased the germination rate and relative emergence index. Generally, growth index, shoot fresh and dry weights, true leaf number, and stem diameter increased as the amount of N increased within each fertilizer type. The best shoot growth was observed in the highest doses of Conventional and Dramatic 2-4-1 treatments (0.84 g/L N). However, Dramatic 2-4-1 treatments resulted in the lowest root growth when compared to other fertilizers at the same N dose. The second highest fertilization dose (0.56 g/L N) of Sustane 4-6-4 had the best root growth according to root fresh weight, root volume, root area, total root length, as well as the numbers of root tip and crossing when compared to other treatments. For seedlings, a well-developed root system can ensure a good seedling establishment and high survival rate under stressful field conditions after transplanting. Thus, Sustane 4-6-4 at 14 g/L (0.56 g/L N) is recommended to produce high-quality organic watermelon seedlings among the treatments applied in this study.

## INTRODUCTION

*Citrullus lanatus*, commonly known as watermelon, is a popular fruit crop that is widely cultivated around the world. It is a trailing or vining plant that can grow up to 5–6 m in

Corresponding author
Genhua Niu,
Genhua.Niu@ag.tamu.edu

length, produces large, juicy, and sweet fruits (*Wehner, 2008*). It is a warm-season crop which is sensitive to both heat and cold conditions and requires temperatures between 21 °C and 32 °C for optimal growth and fruit production (*Bates & Robinson, 1995*; *Korkmaz & Dufault, 2001*). Watermelon prefers soil that is well-drained, fertile, and slightly acidic with a pH range between 6.0 and 7.0, requires full sun exposure, with fruits usually taking about 75 to 120 days to reach maturity, at which point they are ready to harvest (*Welbaum, 2015*). In 2021, global watermelon production reached 101.6 million tons, with China ranking first and accounting for 59.9% of the world total production. The USA contributed about 1% of the global watermelon production output (*FAOSTAT, 2023*). In the USA, watermelon is produced in southern states, with the top four states including Florida, Georgia, Texas, and California accounting for 75% of total production (*USDANASS , 2022*). The demand for watermelon continues to increase due to its popularity as fresh produce, its many health-beneficial properties, and its versatility in both sweet and savory dishes (*Singh et al., 2001*; *Perkins-Veazie, Davis & Collins, 2012*; *Maoto, Beswa & Jideani, 2019*; *Abou El-Goud, 2020*). In 2021, about 37% of the domestic demand for watermelon in the USA was met by importation (*USDANASS , 2022*). The increasing demand for locally produced fresh produce makes organic watermelon a potentially high profitable crop for growers in the USA.

In the USA, organic agriculture is defined as "an ecological production management system that promotes and enhances biodiversity, biological cycles, and soil biological activity" (*USDA-NOSB, 1995*). Organic agriculture has been developing rapidly in recent years, due to higher profits that farmers can earn from organic agricultural products, its environmental friendliness and sustainability, and more opportunities for interactions between farmers and consumers provided by organic agriculture (*Reganold & Wachter, 2016*; *Xue & Shen, 2022*). More and more consumers opt for organic produce for their perceived health and environmental benefits (*Reganold & Wachter, 2016*). According to the Organic Trade Association, organic food sales in the USA alone have grown by nearly 30 billion US dollars in the last decade, in which organic fruits and vegetables accounted for 37% of the total organic food sale in 2021 (*Organic Trade Association, 2022*). As a result of this growing demand, organic agriculture has become an increasingly important aspect of the farming industry. The availability of organic watermelon transplants is one of the fundamental factors to the development of organic watermelon production.

Transplants including seedlings or sprouted vegetative propagation materials are commonly used to produce plants with consistent quality and regulate plant spacing in the field (*Russo, 2005*; *Unal, 2013*). In addition, plant establishment through transplanting exhibits greater tolerance to both biotic and abiotic stresses, resulting in earlier and higher crop yields when compared to direct seeding (*Leskovar & Cantliffe, 1993*; *Khozaei et al., 2020*). Organic transplants are widely needed in the organic production of plants, especially for vegetable and fruit crops (*Pascual et al., 2018*). Also, the demand for organic transplants of fruits and vegetables is growing rapidly, far exceeding the supply (*Barrett, Zhao & Hodges, 2012*; *Gravel, Dorais & Ménard, 2012*; *Burnett, Mattson & Williams, 2016*). Some growers are producing their own organic transplants. However, the quality is low due to the lack of production and management guidance. On one hand, seedlings are

produced in confined cells within a short growth period (3–6 weeks). Mechanical damage of roots is usually the main cause of transplant 'shock' upon transplanting (*Qin & Leskovar, 2020*). A well-developed root system can help to ensure the root ball is pulled out from the cell with the least mechanical damage (*Wei, Wang & Jeong, 2020*). Therefore, a vigorous root system is typically associated with high seedling quality and subsequently high yield after transplanting (*Kerbiriou et al., 2013*). On the other hand, mineral nutrients in organic fertilizers are not immediately available to plants, which makes fertility management difficult for growers (*Pascual et al., 2018*). For example, organic nitrogen (N) needs to be mineralized by microorganisms and converted to nitrate and ammonium to be available to plants (*Pascual et al., 2018*; *Guo et al., 2019*). However, this mineralization is affected by many factors in substrate, including substrate type and particle size, temperature, and moisture (*Updegraff et al., 1995*; *Jagadamma et al., 2014*; *Yu et al., 2022*). The management of organic substrates and organic fertilizers in organic transplant production is still a new field of research, and the determination of the proper fertilization amount and frequency are challenging (*Bhunia et al., 2021*). Organic fertilizer can be derived from animal manures, plant residuals, by-products of agricultural/food production, and dried microbes (*Chatzistathis et al., 2021*; *Bergstrand, 2022*). Different sources of organic fertilizers possess distinct advantages and disadvantages, leading to diverse changes in the physical, chemical, and biological properties of soil. For example, animal manure has been reported to change the soil microbial diversity, decreasing the abundance of soil borne disease caused microbe *Fusarium* and increasing soil nitrification potential, while animal manure also induced heavy metal contamination problems (*Zhong et al., 2010*; *Yang et al., 2017*; *Ding et al., 2017*). The diverse physical and chemical characteristics of organic fertilizers from different sources will have distinct impacts on plant growth and development, demanding further research to optimize their application for specific crops and environments. Additionally, recent studies have recommended applying organic and inorganic fertilizers together to improve nutrient availability for seedling growth, as organic fertilizers release nutrients slowly (*Assefa & Tadesse, 2019*; *Chatzistathis et al., 2021*). However, for certified organic production, only organic fertilizers are allowed to be used in transplant production (*Bergstrand, 2022*). Thus, there is an urgent need for more studies to quantify the effectiveness of organic fertilizers in promoting plant growth and development of transplants. We hypothesized that the effects of different organic fertilizers on plant growth of watermelon seedlings vary, with the optimal application amount differing for each fertilizer. The objective of this study was to investigate the effects of organic fertilizer type and dose on watermelon germination, seedling growth and development, as well as the mineral content both in substrates and leaves.

## MATERIALS & METHODS

### Plant materials and cultivation conditions

Seeds of watermelon (*Citrullus lanatus*) variety Jubilee (Seedway, LLC., Hall, NY, USA) were sown in 72-cell trays, with two seeds per cell. After three days, the seedlings were thinned to one per cell. Two experiments were conducted in a growth chamber at the Texas

A&M AgriLife Research Centers in Dallas, TX, USA, in 2022 and 2023. The photosynthetic photon flux (PPF) was set to 200 $\mu$mol m$^{-2}$ s$^{-1}$ for all treatments using a dimmer at the canopy level. Full-spectrum LED lights provided a photoperiod of 16 h d$^{-1}$ (PhysioSpec Indoor, Fluence, Austin, TX, USA). The first experiment was conducted from 12 Sept. 2022 to 5 Oct. 2022, and seedlings were harvested 23 days after sowing (DAS). The growth chamber temperature was maintained between 27 and 29 °C. The actual temperature was 27.93 ± 0.06 °C (mean ± standard error), relative humidity was 36.93 ± 0.19%. The second experiment (repeating with same treatments) was conducted from Jan. 19, 2023, to Feb. 13, 2023, and seedlings were harvested 25 DAS. The actual temperature was 27.59 ± 0.04 °C, relative humidity was 63.37 ± 0.91%.

## Treatments and experimental design

To investigate the effects of different organic fertilizers on watermelon seedling production, one conventional fertilizer and three organic fertilizers were tested. Four fertilization doses were used for each fertilizer type. Conventional substrate BM2 and certified organic substrate OM2 were used with the conventional fertilizer and organic fertilizers, respectively. The fertilizer treatment design is shown in Table 1. For the Conventional group, conventional substrate (BM2, Berger, Saint-Modeste, QC, Canada) and conventional fertilizer (ICL Peters Professional 20−20−20, Everris, Dublin, OH, USA) were employed. Organic substrate (OM2, Berger) was used for all the organic fertilizer treatments (Sustane 4−6−4, Nature Safe 7−7−7, and Dramatic 2−4−1). The reason to include a conventional treatment was to determine if organically grown transplants could achieve similar or better growth and quality to those of conventionally grown transplants. BM2 and OM2 were produced by the same manufacturer (Berger). Their ingredients both consist of finely graded peat moss, perlite, vermiculite, dolomitic and calcitic limestone and non-ionic wetting agent with the distinction that the ingredients in OM2 are approved by the Organic Materials Review Institute (OMRI). Their physical and chemical properties, and nutrient content are presented in Table S1. The pH, particle size, and water content of BM2 and OM2 are very similar. It needs to be mentioned that in the Conventional group, BM2 contained a starter fertilizer, whereas no starter fertilizer was included in OM2. However, this problem has been effectively addressed by incorporating the starter fertilizer into our applied fertilizer strategy. In other words, the quantity of Peters Professional fertilizers applied was reduced, with the remaining portion being supplemented by the starter fertilizer contained in the BM2 substrate. Conventional fertilizer Peters Professional 20−20−20 contains 20% N, 20% P$_2$O$_5$ (4.37% P), and 20% K$_2$O (16.56% K), deriving from ammonium phosphate, potassium nitrate, urea, potassium phosphate, magnesium sulfate, boric acid, copper EDTA, iron EDTA, manganese EDTA, ammonium molybdate, and zinc EDTA. Three organic fertilizers in different forms were used in this study. Sustane 4-6-4, containing 4% N, 6% P$_2$O$_5$ (1.31% P), and 4% K$_2$O (3.31% K), produced by Sustane Corporate (Cannon Falls, MN, USA), is a slow-release organic fertilizer in granule form. In addition, Sustane contains 5% humate by volume, made from biologically stable compost, natural potash, and feather meal. Nature Safe 7-7-7, containing 7% N, 7% P$_2$O$_5$ (1.53% P), and 7% K$_2$O (5.80% K), produced by Nature Safe Fertilizers (Cold Spring, KY, USA), is a water-soluble

organic fertilizer in dry flowable powder form, derived from corn steep liquor. Dramatic 2-4-1, containing 2% N, 4% $P_2O_5$ (0.87% P), and 1% $K_2O$ (0.83% K), produced by Dramm Corporation (Manitowoc, WI, USA), is an organic fertilizer in liquid form, made from recycled fish scraps, supplemented with kelp. In addition to the N, P, and K contents listed above, Table S2 details the other mineral nutrients present in conventional and organic fertilizers. Except for Sustane $4-6-4$, all fertilizers used in this study needed to be diluted with water before use, and deionized water was used. According to the manufacturer's recommendation, Sustane $4-6-4$ needs to be incorporated into the substrate in advance for seedling production. Therefore, it was only applied once before sowing the seeds. The other fertilizers were applied weekly, at 0 (before sowing the seeds), 7, 14, and 21 DAS, for a total of four applications during the experiment period, as shown in Table 1. In our preliminary trials, applying the water-soluble fertilizers only once at the beginning caused severe mold problems and reduced germination rates at the high N doses. Therefore, in this study, we chose to apply weekly. However, the cumulative N dose added to each fertilizer group (Conventional, Sustane, Nature Safe, and Dramatic) was matched at 7.78 mg/plant, 15.56 mg/plant, 31.12 mg/plant, and 46.68 mg/plant throughout the experiment period, respectively, as shown in Table 1. The experiment followed a randomized complete block design with three replicates. Each experimental unit consisted of 24 watermelon plants for a total of 72 plants per treatment. To optimize growth rates and accommodate potential variations caused by different shelf locations, the plug trays were rearranged on a weekly basis using a randomized approach. The experiment was repeated two times (block over time).

## Data collection
### Germination
Germination data were collected daily between 3 DAS and 21 DAS in the first experiment. The germination rate was calculated using data from the final day of germination as:

$$\text{Germination rate (GR)} = \frac{\text{Number of emerged seedlings}}{\text{Total number of seeds sowed per experimental unit}} \times 100\%.$$

The emergence index (*Niu et al., 2012*) was calculated as:

$$\text{Emergence index} = \left( \frac{GR_3}{3} + \frac{GR_4}{4} + \ldots + \frac{GR_{21}}{21} \right) \times 100\%$$

In the above equation, GR3, GR4, …, GR21 refer to the germination rate (GR) at the third, fourth, …, and 21st day, respectively.

### Plant morphological and growth parameters
At harvest, six plants per experimental unit were randomly collected for plant morphology analysis. Each plant was investigated individually. Hypocotyl length (cm) was measured from the base of the stem to the point of attachment of the cotyledons. Growth index (cm) was calculated as the average plant height (cm), the greatest width (width 1, in cm) and the perpendicular width (width 2, in cm). A digital caliper was used to measure the stem diameter (mm). Leaf area ($cm^2$) was collected using a leaf area meter (LI-3100C, LI-COR Biosciences, Lincoln, NE, USA). The fresh weights (g) of the shoot and root

**Table 1  Fertilizer treatment design in this study.**

| Treatment | Fertilizer | Fertilization dose per volume of substrate | Nitrogen dose per volume of substrate | Cumulative nutrient dose (mg/plant) | | |
|---|---|---|---|---|---|---|
| | | | | N | P | K |
| Conventional 1 | Peters Professional 20−20−20 | 0.153 g/L for 4 times | 0.14 g/L | 7.78 | 1.70 | 6.44 |
| Conventional 2 | Peters Professional 20−20−20 | 0.328 g/L for 4 times | 0.28 g/L | 15.56 | 3.40 | 12.88 |
| Conventional 3 | Peters Professional 20−20−20 | 0.678 g/L for 4 times | 0.56 g/L | 31.12 | 6.80 | 25.76 |
| Conventional 4 | Peters Professional 20−20−20 | 1.028 g/L for 4 times | 0.84 g/L | 46.68 | 10.20 | 38.64 |
| Sustane 1 | Sustane 4−6−4 | 3.5 g/L for 1 time | 0.14 g/L | 7.78 | 2.55 | 6.44 |
| Sustane 2 | Sustane 4−6−4 | 7 g/L for 1 time | 0.28 g/L | 15.56 | 5.10 | 12.88 |
| Sustane 3 | Sustane 4−6−4 | 14 g/L for 1 time | 0.56 g/L | 31.12 | 10.20 | 25.76 |
| Sustane 4 | Sustane 4−6−4 | 21 g/L for 1 time | 0.84 g/L | 46.68 | 15.30 | 38.64 |
| Nature Safe 1 | Nature Safe 7−7−7 | 0.5 g/L for 4 times | 0.14 g/L | 7.78 | 1.70 | 6.44 |
| Nature Safe 2 | Nature Safe 7−7−7 | 1 g/L for 4 times | 0.28 g/L | 15.56 | 3.40 | 12.88 |
| Nature Safe 3 | Nature Safe 7−7−7 | 2 g/L for 4 times | 0.56 g/L | 31.12 | 6.80 | 25.76 |
| Nature Safe 4 | Nature Safe 7−7−7 | 3 g/L for 4 times | 0.84 g/L | 46.68 | 10.20 | 38.64 |
| Dramatic 1 | Dramatic 2−4−1 | 1.75 g/L for 4 times | 0.14 g/L | 7.78 | 3.40 | 3.22 |
| Dramatic 2 | Dramatic 2−4−1 | 3.5 g/L for 4 times | 0.28 g/L | 15.56 | 6.80 | 6.44 |
| Dramatic 3 | Dramatic 2−4−1 | 7 g/L for 4 times | 0.56 g/L | 31.12 | 13.60 | 12.88 |
| Dramatic 4 | Dramatic 2−4−1 | 10.5 g/L for 4 times | 0.84 g/L | 46.68 | 20.40 | 19.32 |

**Notes.**
Cumulative nutrient dose refers to the total amount of nutrients received by each plant throughout the experiment period. N, P, and K refer to nitrogen, phosphorus, and potassium, respectively.

were determined promptly after harvesting and carefully washing off the substrate. Fresh samples were oven-dried at 70 °C until they reached a constant weight to determine the shoot and root dry weights (g). Plant compactness (g/cm) was calculated as shoot dry weight by plant height (*Elkins & van Iersel, 2020*).

### Relative chlorophyll content and chlorophyll fluorescence

On the same day of harvest, three plants were randomly selected from each experimental unit for relative chlorophyll content and chlorophyll fluorescence analyses. Relative leaf chlorophyll content was analyzed by a handheld chlorophyll meter (SPAD-502, Konica Minolta Sensing Inc., Tokyo, Japan), expressed as SPAD, with three measurements on different fully expanded mature leaves in each plant averaged to obtain a single SPAD reading. Leaf chlorophyll fluorescence was measured by a portable chlorophyll fluorimeter (Pocket PEA, Hansatech Instruments Ltd., Norfolk, The United Kingdom). The data were taken from a randomly picked fully expanded mature leaf from each plant after dark adaptation for 20 min. Maximum quantum yield of photosystem II ($F_v/F_m$) and the multi-parametric parameter, photosynthetic performance index ($PI_{abs}$) were calculated using the software PEA Plus (Version 1.13, Hansatech Instruments Ltd.) (*Strasser, Srivastava & Tsimilli-Michael, 2000*; *Hooks, Niu & Ganjegunte, 2019*). Both $F_v/F_m$ and $PI_{abs}$ are sensitive indicators of plant stress, can be used to assess the health and efficiency of the photosynthetic apparatus in plants.

### Root morphology

After measuring the fresh weight, root samples were placed in a transparent methyl methacrylate plate with a small amount of water to spread the root hairs. The sample was then scanned by an Epson scanner (Perfection V850, Epson America. Inc., Long Beach, CA, USA). Six root samples per experimental unit were investigated. Each root sample was investigated individually. Root growth parameters, including root volume ($cm^3$), root area ($cm^2$), average root diameter (mm), total root length (cm), and root tip, fork, and crossing numbers, were analyzed using WinRHIZO software (version 2022b, Regent Instruments Inc., Québec City, Quebec, Canada).

### Mineral nutrient analysis

Dry leaf tissues were ground using a Wiley mill (Thomas Scientific, Swedesboro, NJ) to pass a 40-mesh (0.42 mm) sieve. At least 1 g of dried leaf powder samples and 500 mL dried substrate samples were collected from each experimental unit. Samples of leaf and substrate were sent to Texas A&M AgriLife Extension Service Soil, Water, and Forage Testing Laboratory for mineral analyses. Phosphorus (P), potassium (K), calcium (Ca), magnesium (Mg), sodium (Na), and sulfur (S) in dried substrate samples were extracted using the Mehlich III extractant method and were determined by Inductively Coupled Plasma (ICP) (*Mehlich, 1978*; *Mehlich, 1984*). Copper (Cu), iron (Fe), manganese (Mn), and zinc (Zn) in dried substrate samples were extracted using a 0.005 M diethylenetriaminepentaacetic acid (DTPA), 0.01 M $CaCl_2$, and 0.1 M triethanolamine solution and then analyzed by ICP (*Lindsay & Norvell, 1978*). Boron (B) was extracted from dried substrate sample using a hot-water extraction and then analyzed by ICP (*de Abreu et al., 1994*). Total N content was analyzed after high temperature combustion process (*Sheldrick, 1986*). Minerals in leaf samples were analyzed by ICP after nitric acid digestion (*Isaac & Johnson, 1975*; *Havlin & Soltanpour, 1980*).

### Statistical analyses

All data were analyzed using analysis of variance (ANOVA) with the PROC GLM procedure which uses the method of least squares to fit the general linear models, in SAS software (Version 9.4; SAS Institute Inc., Cary, NC, USA). The specific factors analyzed were fertilizer type and fertilization dose, and the results were expressed as the mean values and standard errors. To ensure the validity of the ANOVA results, graphical diagnostics was performed to visually explore the distribution of the data and identify any potential outliers, normality of the data was checked using Shapiro-Wilk test, and homogeneity of variance was assessed using Levene's test. Tukey's honestly significant difference (HSD) test was performed at a significance level of $\alpha = 0.05$ to further explore significant differences between means of treatment groups identified by ANOVA.

## RESULTS

Two-way ANOVA results of p-values for all parameters are shown in Tables 2–5 and Tables S3–S7 with degree of freedom and F values presenting in the Supplementary Materials.

**Table 2 Two-way ANOVA results of p-values for germination, plant growth, chlorophyll content, and chlorophyll fluorescence parameters.**

|  | Germination rate | Emergence index | Shoot FW | Shoot DW | Shoot water content | Root DW | Root water content | SPAD | Fv/Fm | PI$_{abs}$ |
|---|---|---|---|---|---|---|---|---|---|---|
| Fert | 0.0248 | 0.0171 | <0.0001 | <0.0001 | <0.0001 | <0.0001 | 0.0671 | <0.0001 | <0.0001 | 0.0392 |
| Dose | 0.0442 | 0.0007 | <0.0001 | <0.0001 | <0.0001 | 0.0081 | 0.0725 | <0.0001 | <0.0001 | <0.0001 |
| F × D | 0.0829 | 0.0189 | 0.0009 | 0.2813 | <0.0001 | <0.0001 | 0.0125 | 0.0234 | <0.0001 | 0.0002 |

Notes.
   'Fert' and 'F' refer to the type of fertilizer; 'Dose' and 'D' refer to the nitrogen dose; 'FW' refers to fresh weight; 'DW' refers to dry weight.

**Table 3 Two-way ANOVA results of p-values for shoot morphological and growth parameters.**

|  | Hypocotyl length | Plant height | Growth index | Leaf number | Total leaf area | Root/ shoot | Stem diameter | Compactness |
|---|---|---|---|---|---|---|---|---|
| Fert | <0.0001 | <0.0001 | <0.0001 | <0.0001 | <0.0001 | <0.0001 | <0.0001 | <0.0001 |
| Dose | <0.0001 | <0.0001 | <0.0001 | <0.0001 | <0.0001 | <0.0001 | <0.0001 | 0.5434 |
| F × D | 0.0016 | <0.0001 | 0.1191 | 0.1464 | 0.0057 | 0.8864 | 0.1426 | 0.0060 |

Notes.
   'Fert' and 'F' refer to the type of fertilizer; 'Dose' and 'D' refer to the nitrogen dose; 'Root/shoot' refers to root to shoot dry weight ratio.

**Table 4 Two-way ANOVA results of p-values for root morphology parameters.**

|  | Total root length | Root area | Average root diameter | Root volume | Number of root tips | Number of root crossings |
|---|---|---|---|---|---|---|
| Fert | <0.0001 | <0.0001 | <0.0001 | <0.0001 | <0.0001 | <0.0001 |
| Dose | <0.0001 | 0.0009 | <0.0001 | 0.0360 | 0.0225 | 0.0036 |
| F × D | 0.0206 | 0.0054 | 0.0084 | 0.0012 | 0.3109 | 0.0234 |

Notes.
   'Fert' and 'F' refer to the type of fertilizer; 'Dose' and 'D' refer to the nitrogen dose.

## Germination

Germination rate and emergence index are presented in Fig. 1. Results indicate that the germination rate ranged from 67.3% in Sustane 4 treatment to 88.2% in Conventional 4 treatment. Sustane 4 treatment decreased germination rate when compared with Conventional 2–4, Sustane 1, and Nature Safe 1–2. The emergence index had similar trends, with Sustane 4 treatment having the lowest emergence index of 42.8%.

## Plant morphological and growth parameters

At harvest, Fig. 2 shows representative shoot and root photos of watermelon seedlings of each treatment. The photos provide a visual impression of the plants under different treatments.

Hypocotyl length, plant height, growth index, true leaf number, total leaf area, root to shoot DW ratio, stem diameter, and compactness are shown in Fig. 3. Hypocotyl length ranged from 6.81 cm in Nature Safe 1 treatment to 8.49 cm in Dramatic 4 treatment (Fig. 3A). Plant height ranged from 11.8 cm to 33.9 cm (Fig. 3B). Dramatic 4 treatment resulted in the highest plant height. The difference between the Conventional 4 and Dramatic 3

**Table 5 Macronutrient contents in substrate and leaf of watermelon seedlings under different fertilizer treatments.**

| Ttreatment | Substrate N mg/kg | Substrate P mg/kg | Substrate K mg/kg | Leaf N % | Leaf P mg/kg | Leaf K mg/kg |
|---|---|---|---|---|---|---|
| Conventional 1 | 5 e | 18 g | 25 g | 2.89 fgh | 5.76 e | 23.4 def |
| Conventional 2 | 27 cde | 50 fg | 41 fg | 4.39 cd | 10.70 de | 26.7 bcde |
| Conventional 3 | 66 b | 109 cd | 135 bc | 5.92 ab | 13.52 cde | 31.5 ab |
| Conventional 4 | 131 a | 167 b | 254 a | 6.58 a | 17.73 bcde | 35.0 a |
| Sustane 1 | 1 e | 18 g | 36 fg | 1.67 i | 6.39 e | 21.9 ef |
| Sustane 2 | 2 e | 21 g | 47 efg | 2.07 hi | 7.27 e | 24.0 cdef |
| Sustane 3 | 3 e | 35 fg | 88 cdef | 2.79 fgh | 11.74 de | 27.9 bcd |
| Sustane 4 | 20 de | 81 def | 185 b | 3.30 efg | 9.80 de | 29.2 bc |
| Nature Safe 1 | 3 e | 22 g | 34 fg | 2.32 ghi | 7.32 de | 18.5 fg |
| Nature Safe 2 | 1 e | 50 fg | 34 fg | 2.93 fgh | 13.11 cde | 22.2 ef |
| Nature Safe 3 | 20 de | 147 bc | 113 cd | 3.93 de | 26.65 abc | 27.9 bcd |
| Nature Safe 4 | 24 de | 187 b | 133 bc | 4.57 cd | 34.02 a | 28.4 bcd |
| Dramatic 1 | 2 e | 60 efg | 16 g | 2.59 ghi | 10.04 de | 15.8 g |
| Dramatic 2 | 9 e | 107 cde | 16 g | 3.71 def | 15.82 bcde | 18.7 fg |
| Dramatic 3 | 62 bc | 187 b | 64 defg | 5.20 bc | 28.47 ab | 23.2 def |
| Dramatic 4 | 57 bcd | 428 a | 105 cde | 6.21 a | 21.51 abcd | 26.6 bcde |
| Sufficiency range[*] | | | | 2.5–4.5 | 2–7.5 | 15–55 |

| | | Substrate N mg/kg | Substrate P mg/kg | Substrate K mg/kg | Leaf N % | Leaf P mg/kg | Leaf K mg/kg |
|---|---|---|---|---|---|---|---|
| P | Fert | <0.0001 | <0.0001 | <0.0001 | <0.0001 | <0.0001 | <0.0001 |
| | Dose | <0.0001 | <0.0001 | <0.0001 | <0.0001 | <0.0001 | <0.0001 |
| | Fert × D | <0.0001 | <0.0001 | <0.0001 | <0.0001 | <0.0001 | 0.6208 |
| F | Fert | 42.36 | 197.74 | 19.31 | 135.47 | 41.37 | 39.30 |
| | Dose | 52.64 | 314.32 | 121.28 | 172.17 | 57.94 | 71.12 |
| | Fert × D | 10.92 | 41.19 | 6.37 | 6.05 | 8.19 | 0.80 |

**Notes.**

[*]Data source: Approximate sufficiency ranges of the minerals in mature leaf tissue (*Kalra, 1997*).

'Fert' refers to the fertilizer type. 'Dose' and 'D' refer to the nitrogen dose.

Different letters within one column suggest significant differences among fertilizer treatments indicated by Tukey's honestly significant difference test at $P < 0.05$.

The last six rows show the two-way ANOVA test results of $p$-values and $F$ values.

treatments is not significant. Conventional 1, Sustane 1 and 2, Nature Safe 1 and 2, and Dramatic 1 treatments had comparable lowest plant height ranging from 11.8 cm to 16.2 cm. Growth index ranged from 9.4 cm in Sustane 1 to 19.9 cm in Dramatic 4 (Fig. 3C). Similar to plant height, Dramatic 4 showed the highest growth index. In comparison, Sustane 1 treatment showed the lowest growth index. True leaf number and total leaf area shared similar trends among fertilizer treatments with growth index (Figs. 3C–3E). Dramatic 4 had the highest true leaf number of 7.92 and total leaf area of 185.6 cm$^2$, while Sustane 1 showed the lowest true leaf number of 2.58 and total leaf area of 18.8 cm$^2$ among all the fertilizer treatments. The root to shoot DW ratio ranged from 3.5% to 18.3% (Fig. 3F). The highest root to shoot DW ratios were observed in the Sustane 1 and Nature Safe 1 treatments, with values of 17.5% and 18.3%, respectively. In contrast, the lowest root to shoot DW ratios were found in the Dramatic 3 and Dramatic 4 treatments, measuring 4.77% and 3.48% respectively. On average, the lowest root to shoot DW ratios
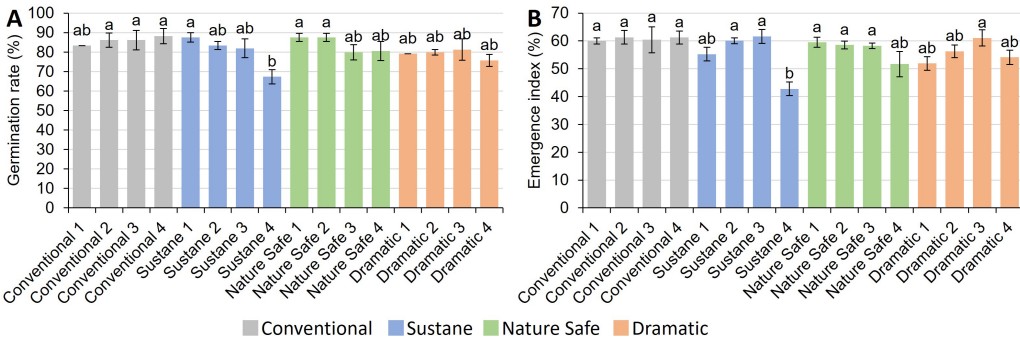

**Figure 1 Germination rate and emergence index of watermelon seedlings under different fertilizer treatments.** (A) Germination rate (%); (B) emergence index (%). Different letters on the top of each bar indicate significant differences among 16 fertilizer treatments tested by Tukey's honestly significant difference test at $P < 0.05$. Results were expressed as mean ± standard error (SE).

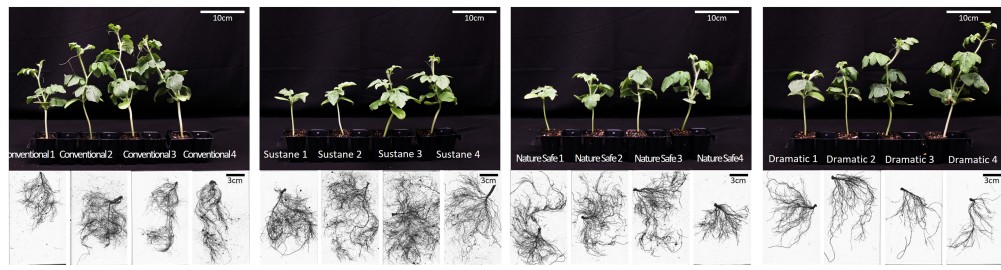

**Figure 2 Shoot photo and root morphology of watermelon seedlings at harvest-time.** .

were approximately 4.34 times lower than the highest root to shoot DW ratio group. The highest stem diameters were found in Conventional 3, Conventional 4, and Dramatic 4 treatments, ranged from 3.98 mm to 4.06 mm, with no significant difference between treatments (Fig. 3G). Compactness ranged from 1.75 g/m in Nature Safe 1 to 2.46 g/m in Sustane 2 (Fig. 3H). There was no significant difference in compactness among different N doses within the same fertilizer treatment. However, Sustane 1 increased compactness when compared to Nature Safe 1 at a N dose of 0.14 g/L. Sustane 2 also showed significantly higher compactness than Conventional 2 at a N dose of 0.28 g/L.

Fresh weight (FW), dry weight (DW), water content of shoot, DW and water content of root, relative chlorophyll content (SPAD), and leaf chlorophyll fluorescence parameters ($F_v/F_m$ and $PI_{abs}$) are presented in Fig. 4. The results showed that shoot FW of watermelon seedlings ranged from 1.82 g in Sustane 1 treatment to 9.48 g in Dramatic 4 treatment. The Dramatic 4 and Conventional 4 treatments exhibited the highest shoot FW, surpassing the second highest shoot FW group consisting of Conventional 3, Nature Safe 4, and Dramatic 3 treatments by averagely 1.27 times (Fig. 4A). Moreover, increasing fertilization dose within the Conventional and Dramatic fertilizer groups led to significant increases in shoot FW. Shoot DW shared a similar trend with shoot FW among the different fertilizer treatments, ranging from 0.23 g in Nature Safe 1 treatment to 0.80 g in Dramatic 4 and Conventional

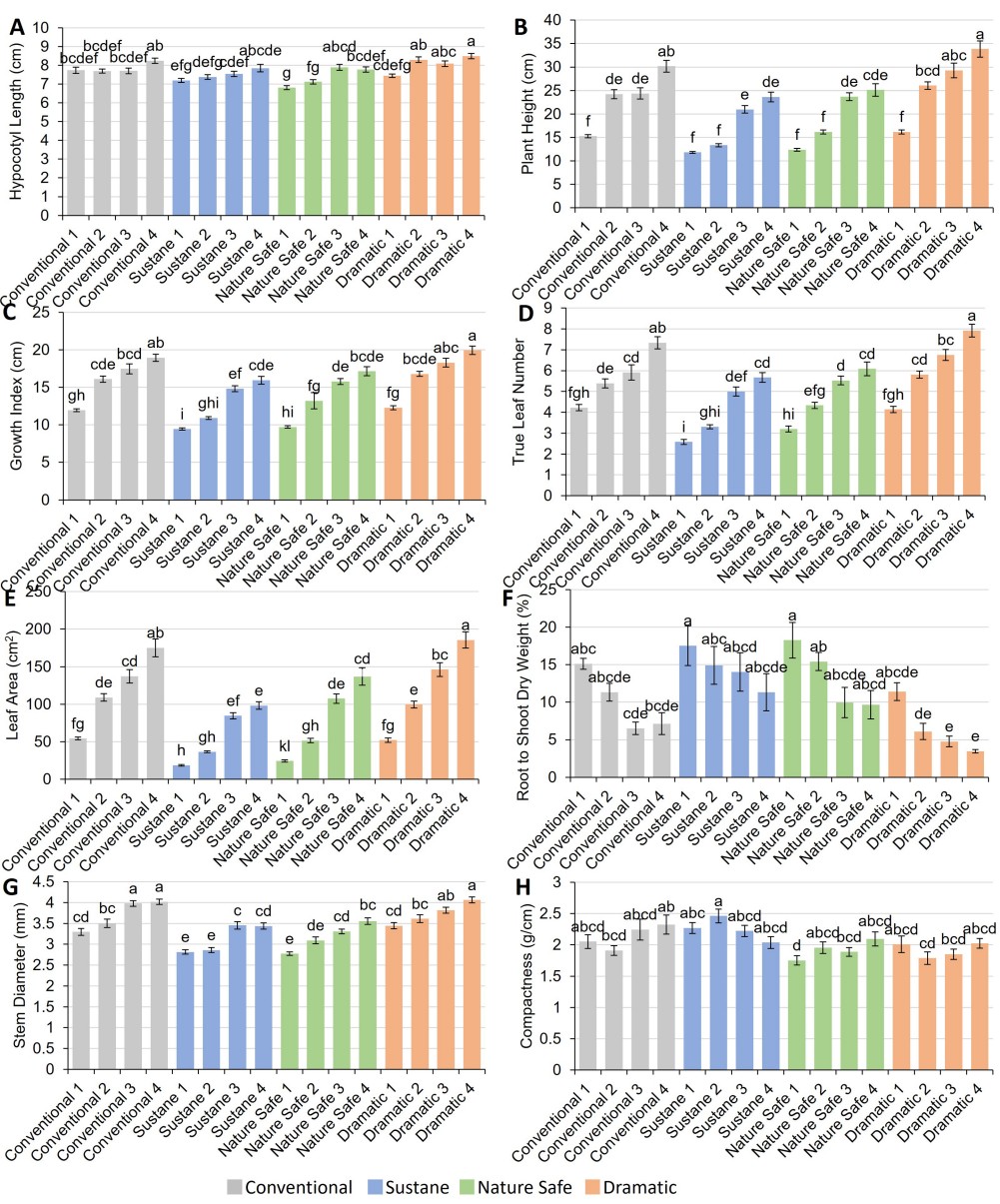

**Figure 3 Plant shoot morphological parameters of watermelon seedlings under different fertilizer treatments.** (A) Hypocotyl length (cm); (B) plant height (cm); (C) growth index (cm); (D) true leaf number; (E) leaf area (cm$^2$); (F) root to shoot dry weight ratio (%); (G) stem diameter (mm); (H) compactness (g/cm). Different letters on the top of each bar indicate significant differences among 16 fertilizer treatments tested by Tukey's honestly significant difference test at $P < 0.05$. Results were expressed as mean ± SE.

4 treatments (Fig. 4B). The highest shoot water content was found in Conventional 3 and 4 of 92.5% and 92.4%, respectively (Fig. 4C). The lowest shoot water content was found in Sustane 1 treatment of 86.0%. Interestingly, despite Dramatic 3 and 4 showing higher shoot FW and DW when compared with other organic fertilizers at the same N dose, Dramatic

3 and 4 treatments had the lowest and second lowest root DW of 0.025 g and 0.027 g, respectively (Figs. 4A, 4B, and 4D). The highest root DW was found in Conventional 2 treatment of 0.05 g, which was about 2 times higher than the lowest root DW found in Dramatic 3 treatment (Fig. 4D). The root water content across different treatments varied between 92.8% and 95.0% (Fig. 4E). Among these treatments, Conventional 3, Dramatic 2, and Dramatic 4 equally exhibited the highest levels of root water content. However, the Nature Safe 4 treatment had the lowest root water content of 92.8%, significantly lower than the treatments in the highest group.

### Relative chlorophyll content and chlorophyll florescence

Relative chlorophyll content (SPAD) ranged from 36.3 to 49.2, with Conventional 4 and Dramatic 4 treatments showing the highest SPAD while Sustane 1, Sustane 2, and Nature Safe 1 treatments showing the lowest SPAD (Fig. 4F). $F_v/F_m$ ranged from 0.75 in Sustane 1 to 0.80 in Dramatic 4 (Fig. 4G). Performance index ($PI_{abs}$) ranged from 0.46 to 1.96 among treatments, with Conventional 4 and Dramatic 4 showing comparably the highest $PI_{abs}$ (Fig. 4H).

### Root morphology

Total root length ranged from 275 cm to 1,164 cm among treatments (Fig. 5A). The highest total root length was observed in Sustane 3 treatment, while the lowest total root length was found in Dramatic 4 treatment. Within the same fertilizer group, the highest fertilization dose resulted in significantly or insignificantly lower total root length when compared with other fertilization rates. Root volume and root area followed similar trends as total root length among treatments, ranging from 10.8 to 40.3 $cm^2$, 0.33 to 1.11 $cm^3$, respectively (Figs. 5B and 5D). Average root diameter ranged from 0.33 mm in Sustane 2 treatment to 0.40 mm in Dramatic 4 treatment (Fig. 5C). The number of root tips and crossings shared similar trends among treatments, ranging from 979 to 8,581, 236 to 2,538, respectively (Figs. 5E and 5F). For both numbers of root tips and root crossings, the highest value was found in the Sustane 3 treatment, the lowest value was found in the Dramatic 4 treatment.

### Macronutrients in substrate

Contents of N, P, and K in the substrate at harvest time are presented in Table 5. N was highest in Conventional 4 at 131 mg/kg. Results of other minerals are included in the supplementary materials (Table S6). Conventional 1, Sustane 1, Sustane 2, Sustane 3, Nature Safe 1, Nature Safe 2, Dramatic 1, and Dramatic 2 had the lowest N content in substrate, with values below 10 mg/kg. The N content in the substrate was not significantly different between the conventional and organic fertilizer treatments at 0.14 ang 0.28 g/L N dose (Treatment level 1 and 2 in each fertilizer group). The P content of the substrate ranged from 18 mg/kg to 428 mg/kg. The Dramatic 4 treatment showed the highest P content. At 0.14, 0.28, and 0.84 g/L N doses, Dramatic treatment had significantly higher P content in the substrate compared to the other fertilizers. At 0.56 g/L N dose, the P content in the Dramatic treatment was significantly higher than in the Conventional and Sustane treatments. The K content of the substrate was highest in Conventional 4 treatment at 254 mg/kg, and lowest in Dramatic 1 and Dramatic 2 treatments at 16 mg/kg. The K content in

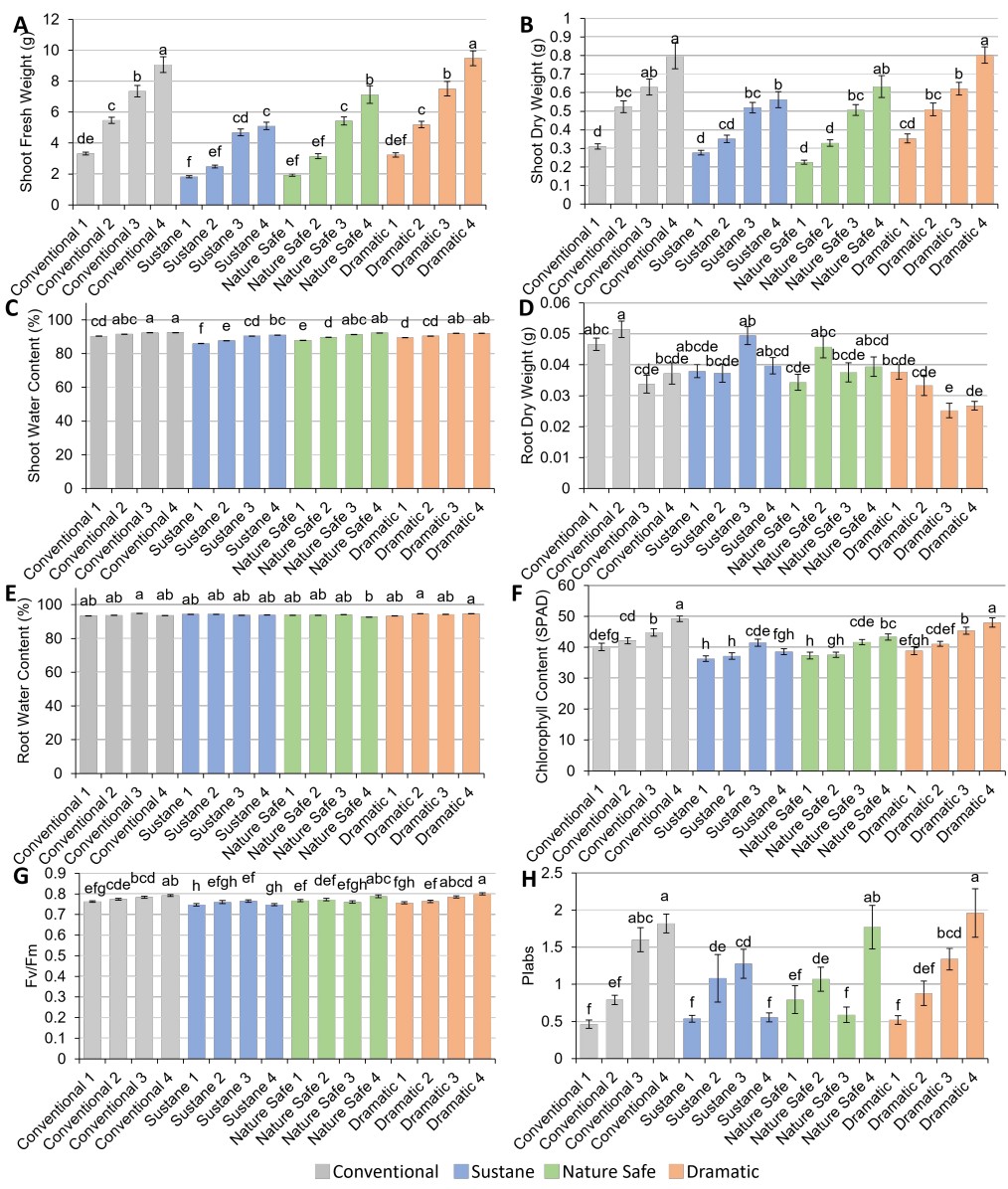

**Figure 4 Plant growth, chlorophyll content and chlorophyll fluorescence of watermelon seedlings under different fertilizer treatments.** (A) Shoot fresh weight (g); (B) shoot dry weight (g); (C) shoot water content (%); (D) root dry weight (g); (E) root water content (%); (F) relative chlorophyll content (SPAD); (G) $F_v/F_m$, the maximum quantum yield of photosystem II; (H) performance index (PI$_{abs}$). Different letters on the top of each bar indicate significant differences among 16 fertilizer treatments tested by Tukey's honestly significant difference test at $P < 0.05$. Results were expressed as mean ± SE.

the substrate was not significantly different among the different fertilizer treatments at N doses of 0.14 and 0.28 g/L. However, at 0.56 g/L N, the Dramatic 3 treatment significantly decreased the K content in substrate by 53% when compared to the Conventional 3 treatment. At 0.84 g/L N, the Dramatic 4 treatment significantly decreased the K content in

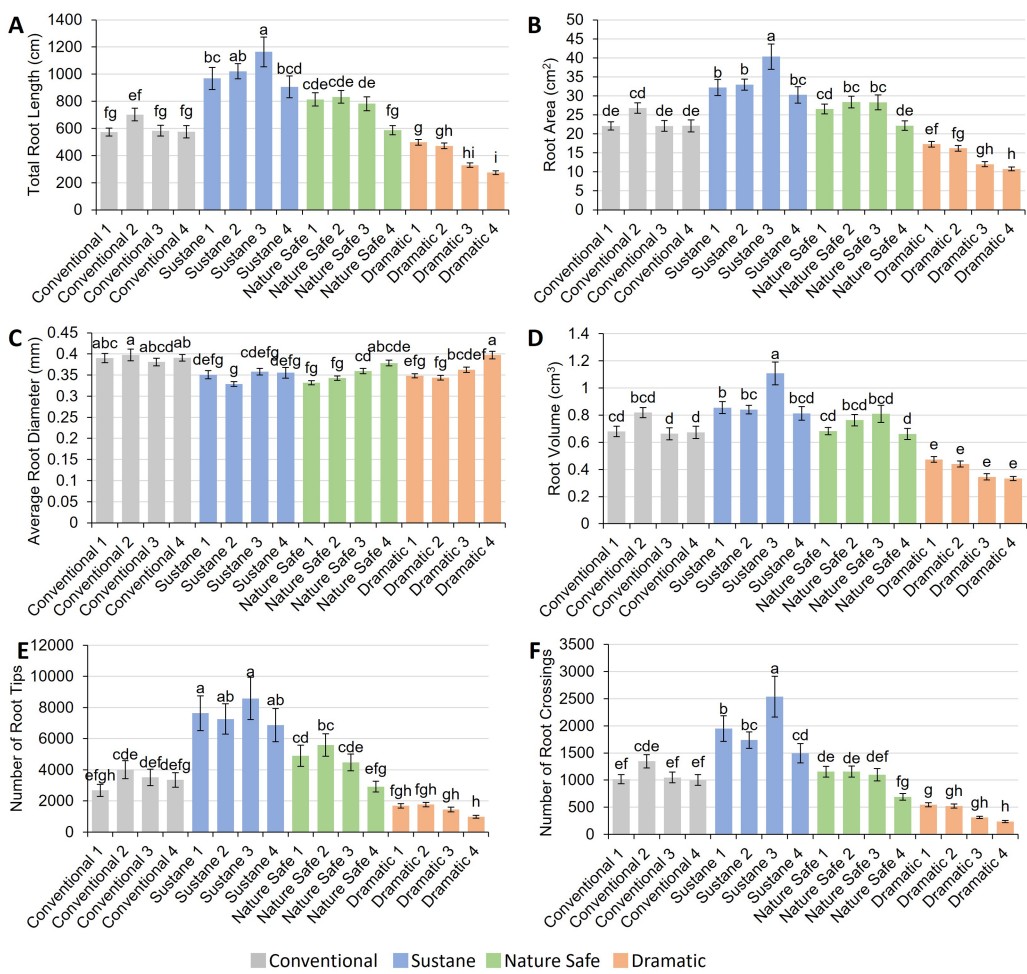

**Figure 5** **Root morphology of watermelon seedlings under different fertilizer treatments.** (A) Total root length (cm); (B) root area (cm$^2$); (C) average root diameter (mm); (D) root volume (cm$^3$); (E) number of root tips; (F) number of root crossings. Different letters on the top of each bar indicate significant differences among 16 fertilizer treatments tested by Tukey's honestly significant difference test at $P < 0.05$. Results were expressed as mean $\pm$ SE.

substrate by 59% and 43% when compared to Conventional 4 and Sustane 4 treatments, respectively.

## Macronutrients in leaf

Contents of N, P, and K in dried leaf tissues of watermelon seedlings at harvest time are shown in Table 5 as well. Results of other minerals are included in supplementary materials (Table S7). The N content in the leaves ranged from 1.67% (w/w) in Sustane 1 treatment to 6.58% (w/w) in Conventional 4 treatment. Leaf N contents were insufficient under Sustane 1, Sustane 2, and Nature Safe 1 treatments; while leaf N contents were over the sufficiency range under Conventional 3, Conventional 4, Nature Safe 4, Dramatic 3, and Dramatic 4 treatments (*Kalra, 1997*). The leaf P content was highest in Nature Safe 4 treatment at 34.0 g/kg, and lowest in Conventional 1 treatment at 5.76 g/kg. The leaf P content was

not affected by the type of fertilizer used at N doses of 0.14 and 0.28 g/L. However, at 0.56 g/L N and 0.84 g/L N, Nature Safe or Dramatic resulted in higher leaf P content than Conventional or Sustane treatments. Leaf P content was sufficient for all treatments, but leaf P content was over the sufficiency range for higher fertilization doses. The leaf K content was highest in Conventional and lowest in Dramatic at each N dose. The highest value found in Conventional 4 treatment, with a leaf K content of 35.0 g/kg. The lowest value was found in Dramatic 1 treatment, with a leaf K content of 15.8 g/kg. Leaf K content under all treatments falls within the sufficiency range.

## DISCUSSION

Organic seedling production can pose more challenges than conventional seedling production due to the complexity of organic substrate and fertilizer management (*Burnett, Mattson & Williams, 2016*). Synthetic fertilizers are highly soluble and are immediately available to seedlings, with fertility management well documented (*Miao, Stewart & Zhang, 2011*; *Mzibra et al., 2021*). On the other hand, organic fertilizers which are made from plant, animal, and mineral materials, first need to be broken down by microorganisms into its mineral components before they can be taken up by plants (*Pascual et al., 2018*). A previous study has shown that increasing fertilization doses of both organic and conventional fertilizers generally can improve the growth of watermelon plants, which is consistent with our results. However, the authors found that very high levels of organic fertilizers can decrease leaf area index and vine length, which is likely due to the increased microbial activity in the soil, leading to nutrient competition and the production of phytotoxic compounds (*Audi, Aguyoh & Gao-Qiong, 2013*). Therefore, providing accurate information regarding the specific type and fertilization dose of organic fertilizers can greatly benefit growers by saving them both money and time while promoting sustainable agricultural practices in seedling production. In this study, we observed that the four fertilization doses had different effects on watermelon seedlings depending on the type of organic fertilizers.

One of the challenges in producing organic seedlings is the low germination rate, which is often caused by the high fertilization dose and/or poor physical properties of the organic substrate such as compaction (*Rivera et al., 2022*). In the current study, the Sustane 4 treatment resulted in a notable decrease in germination rate, specifically by 23.6% compared to the Conventional 4 treatment. Under the Sustane fertilizer treatment, which contains 5% humate by volume, we observed a higher degree of substrate compaction compared to other treatments. This could be attributed to the partial insolubility of Sustane, as some of its humate components remain undissolved in water. For example, humin, a major humate constituent, possesses a large molecular weight ranging from 100,000 to 10,000,000 and exhibits water insolubility at all pH levels (*Pettit, 2004*). These insoluble humin particles likely contributed to the increased substrate density observed in the Sustane-treated group. Increased compaction resulted in low aeration in the substrate which decreased the seedling emergence of soybean (*Hyatt et al., 2007*; *Wolken et al., 2010*). Therefore, the low germination rate observed in the Sustane 4 treatment may have been

due to compaction of the substrate. A higher emergence index indicates a better and faster germination (*Kader, 2005*). Within each group of organic fertilizer treatments, the highest fertilization dose (0.84 g/L N) resulted in low emergence indexes, though significant differences were only observed in Sustane treatments (Fig. 1B). The germination index results corroborate previous studies on eight semi-arid grassland species, which found that higher N doses delayed seed germination (*Zhang et al., 2020*). Our unpublished preliminary experiment (data not shown) showed that applying water-soluble fertilizers once, instead of four times, at the same total N dose, significantly reduced pepper seed germination and caused severe mold problems during early germination. This is why we chose to apply water-soluble fertilizers weekly, rather than all at once.

Chlorophyll absorbs light energy and uses it to power photosynthesis, which is essential for plant growth (*Bannari et al., 2007*). Thus, an increase in relative chlorophyll content (SPAD) corresponds with an increase in shoot biomass in watermelon plants (*Ulas, 2022*). *Colla et al. (2011)* found that SPAD was highly affected by N availability in the substrate. Our results in the Conventional, Nature Safe, and Dramatic groups (Fig. 4A) also support this statement, and a strong correlation between SPAD and leaf N content was observed (Pearson correlation coefficient of 0.89, $P <.0001$). SPAD value in Sustane 3 was lower than that in Sustane 4, possibly due to the stress as reflected in lower $F_v/F_m$ and $PI_{abs}$ (*Niu, Rodriguez & Aguiniga, 2008*; *Banks, 2018*). $F_v/F_m$ and $PI_{abs}$ decreased by 2.2% and 56.7%, respectively, in the Sustane 4 treatment compared to the Sustane 3 treatment. Although $F_v/F_m$ is more frequently used in abiotic stress studies, our results agree with a previous study that $F_v/F_m$ was less sensitive for measuring plant stress under minor abiotic conditions such as drought (*Banks, 2018*). A study of the effects of light spectra on grafted watermelon seedlings found that $PI_{abs}$ can sensitively reflect different abiotic stress factors encountered by the plants, and concluded that $PI_{abs}$ explained the changes in plant biomass among the treatments (*Bantis et al., 2021*). Therefore, the Sustane 4 treatment caused more stress on watermelon seedlings than the Sustane 3 treatment at the time point when the chlorophyll florescence was investigated. This stress caused the inhibition of photosystem II activity, which limited electron transport efficiency. However, it needs to be mentioned that the chlorophyll florescence was only measured once before the harvest, and it can only reflect the on-time stress of the plant. In future studies, chlorophyll florescence is recommended to measure more frequently throughout the plant growth period to get more convincing results.

According to results obtained from watermelon seedling growth and morphology, for all fertilizers, higher fertilization dose increased shoot FW, shoot DW, water content, plant height, growth index, true leaf number, total leaf area, and stem diameter. These results were in line with previous studies on watermelon, melon, dragon spruce, and lettuce, in matured plants or seedlings (*Olaniyi, 2008*; *Zhao & Liu, 2009*; *Castellanos et al., 2011*; *Liu et al., 2014*). However, for some fertilizers, the second highest fertilization dose (0.56 g/L N) and the highest fertilization dose (0.84 g/L N) did not have a significant effect on these parameters. Our results agreed with the existing literature that above a certain N dose, with the increasing of N doses, there is no significant increase on the shoot biomass in mini-watermelon plants (*Colla et al., 2011*). Similar results were also observed in seedlings
of other species, such as blue gum and painted maple (*Razaq et al., 2017*; *Acevedo et al., 2021*). Those studies suggested that the optimal fertilization amount is reached when no more yield increases or improvements in seedling/plant performance with further increases in N dose.

In addition to the N dose, appropriate ratios of N, P, and K in organic fertilizers are important to ensure healthy growth of both shoots and roots. In this study, watermelon seedlings in Dramatic fertilizer treatments exhibited a lower root to shoot DW ratio and a poor root system, especially at higher N doses, compared to plants in other fertilizer treatments, possibly due to the high P and low K ratios (Table 1). When the N dose was matched, the P doses in Conventional and Nature Safe were equal. The K doses in Conventional, Sustane, and Nature Safe were also equal. Sustane and Dramatic had 50% and 100% higher P doses than Conventional and Nature Safe, respectively. Dramatic had a 50% lower K dose than the other three fertilizers. The mineral analysis of substrate and leaf suggested that Dramatic had the lowest K compared to other fertilizers (Table 5). It was reported that low K doses resulted in low root weight in watermelon seedlings (*Pan et al., 2012*). A study on *Arabidopsis* found that K deficiency resulted in a lower proportion of root weight to total weight (root + shoot) (*Hermans et al., 2006*). Therefore, the low K dose might lead to the low root to shoot DW ratio in Dramatic fertilizer treatments at the same N dose (Fig. 2G). On the other hand, high P dose can lead to decreased lateral root growth and increased primary root growth (*Niu et al., 2013*). We found the substrate P content was highest in Dramatic treatments since the P dose was highest in Dramatic among all the fertilizers. Our results on root morphology supported this assumption, as we found that the average root diameter was highest and the number of root tips and crossings was lowest in Dramatic 4 treatment (Figs. 2 and 5), indicating that there was a higher proportion of primary roots and a lower proportion of lateral roots.

Root morphology suggested that Sustane had higher root volume, root area, total root length, and number of root tips and crossings when compared to other fertilizers under the same N dose (Fig. 5), which may be related to the humate contained in Sustane fertilizer. Humate has been found to promote root growth in radish, pea, and corn plants (*Ertani et al., 2011*; *Vasileva & Ilieva, 2015*; *Barzegar et al., 2022*). Additionally, humate promoted lateral root growth in lettuce and corn (*Canellas et al., 2002*; *Busato et al., 2018*). Our study also found that Sustane promoted hair root growth when compared to other fertilizers, which was reflected in root scan pictures (Fig. 2), showing an increase in root tips and crossings numbers (Fig. 5). A superior containerized transplant should have key attributes including compactness, a stout non-elongated stem, and a well-developed balanced root and shoot system (*Leskovar, 2020*). As per this definition, based on the shoot and root weight, morphology data, as well as visual quality, Sustane 3 treatment resulted in higher-quality watermelon seedlings than other treatments. However, according to the mineral analysis (Table 5), substrate N content under Sustane 3 treatment is too low at harvest. To solve this problem, a combination of different organic fertilizers, such as applying Sustane at the start and supplementing water-soluble organic fertilizers during the seedling growth stage, may be an accessible way to produce better organic watermelon seedlings. In future studies, the combined application of organic fertilizers could also benefit the

fertilizer management of Dramatic. We hypothesized that the low K content in Dramatic fertilizer might result in a less developed root system compared to other treatments. Due to the limited sources of organic fertilizers, it is not as easy to adjust the balance of macronutrients and micronutrients as it is with conventional fertilizers (*Chatzistathis et al., 2021*). By combining different organic fertilizers, we can ensure that the fertilizer management meets the requirements of organic production while also achieving a better balance of the different minerals available to the plants.

Mineral analysis of the substrate and leaves under different fertilizer treatments (Table 5, Tables S6, and S7) revealed that some micronutrient contents were higher in organic fertilizers, confirming the trends shown in mineral analysis of fertilizers as shown in Table S2. Fe, Zn, Mn, and Cu content in the substrate under Sustane treatment was generally higher than those under other fertilizer treatments, especially at the highest fertilization dose (0.84 g/L N), confirmed Sustane contains high levels of these micronutrients. However, the differences in micronutrient content among fertilizers in leaves were not as significant as those in the substrate, which implies that micronutrients in organic fertilizers may not be absorbed efficiently by plants. For example, although substrate Fe content was low under Conventional treatments but high in organic fertilizer treatments, leaf Fe content showed opposite trends as in all organic fertilizer treatments were insufficient, indicating the Fe element in organic fertilizers were not as available to the watermelon plants as the Fe in conventional fertilizer. Heavy metal such as Zn and Cu concentrations in organic fertilizers, especially in manure compost, has always been a problem (*Manios, Stentiford & Millner, 2003*; *Lopes et al., 2011*; *Yang et al., 2017*). However, according to results of leaf mineral content, the amount of these micronutrients in watermelon seedling leaves were limited. The absorption of specific metal elements by plants is species dependent (*Uchimiya et al., 2020*). Whether the heavy metal contained in organic fertilizers affects the watermelon fruit needs further studies. Another notable trend was the high Na content under Dramatic treatments. Dramatic fertilizer was made from fish scraps with high Na content (Table S2), and Na content in both the substrate and leaves treated with Dramatic was significantly higher than those treated with other fertilizers, except at the lowest fertilization dose (0.14 g/L N) (Tables S6 and S7). Thus, care needs to be taken when selecting or applying organic fertilizers with high Na salt.

## CONCLUSIONS

This study compared the effects of three organic fertilizers at various N doses to a conventional inorganic fertilizer on watermelon seedling growth, morphology, and physiological characteristics. We observed that, except for the highest dose (0.84 g/L N) of Sustane fertilizer, which negatively impacted germination performance, different fertilization doses of organic fertilizers did not significantly affect watermelon seed germination when compared to the conventional fertilizer group. Although high fertilization doses (0.56 and 0.84 g/L N) of Dramatic resulted in higher performance on shoot growth, they resulted in lower root growth when compared with other fertilizers. Sustane promoted root growth, particularly lateral and hair root growth, suggesting it

may be a good choice for organic watermelon seedling production due to the importance of strong roots for fast establishment after transplanting. The findings of this study are based on the experimental conditions of this study. Results in other conditions might be different. Further research is needed to explore the effectiveness of integrated utilization of different organic fertilizers for producing healthy and vigorous organic seedlings, as well as to evaluate the subsequent performance of watermelon plants after transplanting to field conditions, to determine the optimal organic fertilizer management method based on the crop yield and other relevant factors.

## ACKNOWLEDGEMENTS

We appreciate the assistance of technician Alexandra Spatter with data collection. We also appreciate Dr. Youping Sun, from Utah State University, for his critical review and comments.

### Funding

This study is supported by a Specialty Crop Multi-State Program grant TX-SCM-21-05, Hatch Project TEX07726, and Specialty Crop Block Grant GSC2022030. The funders had no role in study design, data collection and analysis, decision to publish, or preparation of the manuscript.

### Grant Disclosures

The following grant information was disclosed by the authors:
Specialty Crop Multi-State Program: TX-SCM-21-05.
Hatch Project: TEX07726.
Specialty Crop Block Grant: GSC2022030.

### Competing Interests

Qianwen Zhang is an Academic Editor for PeerJ.

### Author Contributions

- Qianwen Zhang conceived and designed the experiments, performed the experiments, analyzed the data, prepared figures and/or tables, authored or reviewed drafts of the article, and approved the final draft.
- Joseph Masabni conceived and designed the experiments, authored or reviewed drafts of the article, and approved the final draft.
- Genhua Niu conceived and designed the experiments, authored or reviewed drafts of the article, and approved the final draft.

### Data Availability

The inspection reports of BM2 and OM2 substrates and mineral contents of fertilizer, substrate and leaf samples and the raw data of the figures are available in the Supplementary File.

## Supplemental Information

Supplemental information for this article can be found online at http://dx.doi.org/10.7717/peerj.16902#supplemental-information.

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
