# Peer review of "Organic fertilizer type and dose affect growth, morphological and physiological parameters, and mineral nutrition of watermelon seedlings"

_PeerJ, doi:10.7717/peerj.16902_

## Round 0.1 · original submission · Major Revisions

Dear Authors, I propose a major revision as I agree with most of the comments of reviewer #2. I believe it is mandatory to respond to the following critical issues: 1) add information on the nutritional composition of the organic fertilizers; 2) please mitigate your conclusion since your study focused on the effects only at the seedling level and therefore it is difficult to find out the best fertilizer; 3) please complete the M&M section and all the missing parts according to Revs' suggestions; 4) comparative statistics should be added; 5) the manuscript will benefit on data that goes more in-depth to the crop throughout its cycle.

Reviewer 1 ·

Basic reporting

N/A

Experimental design

N/A

Validity of the findings

see my additional comments.

Additional comments

The manuscript presents the results of a comprehensive study that investigated the impact of three organic fertilizer types and conventional fertilizers on various growth parameters related to watermelon seedling growth. The authors used a two-way ANOVA analysis to assess the significance of interactions between fertilizer type and growth parameters. They found that, in terms of germination, organic fertilizer treatments were generally comparable to the conventional fertilizer, with an exception at the highest nitrogen level of Sustane 4-6-4. As the nitrogen content increased within each fertilizer type, there was a consistent trend of improved shoot growth parameters. Notably, the study revealed that the second-highest nitrogen level of one organic fertilizer, Sustane 4-6-4, yielded the best root growth, which is critical for seedling establishment and survival in challenging field conditions. As a result, Sustane 4-6-4 at the specified nitrogen level (0.56 g/L N) is recommended for producing high-quality organic watermelon seedlings, emphasizing the importance of tailored organic fertilizer choices in sustainable agriculture. I believe this study is a great example in watermelon organic farming practice and hope the authors can address the following questions before publications.



Major Revisions Suggested:

In the Method sections line 112-123, It is generally a good practice to measure soil contents before conducting fertilizer experiments. In fertilizer treatment experiments, measuring soil nutrient content before and after experiments would help the authors assess the impact of fertilizers on soil fertility, nutrient availability, and the potential for nutrient runoff. This information is crucial to assess growth parameters while minimizing negative environmental impacts. I understand it is very hard to provide the soil content data after treatment but at least suggest the authors provide this piece of data before the treatment in the soil and also discuss the limitations in the discussion part.

Reviewer 2 ·

Basic reporting

In my opinion, the present comparative study of the effectiveness of different fertilizers with a conventional one in the response in growth and physiological parameters in melon is interesting. However, I consider that it is a descriptive study and from which it is difficult to draw clear conclusions with the information provided. Firstly, the nutritional composition of the organic fertilizers applied is not indicated, only their nature is indicated. In this way, it is difficult or impossible to know if the observed effects on growth are due solely to the N content, which is obviously not the case. Furthermore, when studying the effects only at the seedling level, it is difficult to know which fertilizer has the best effectiveness. Strange results are observed, such as, for example, the organic fertilizer "Dramatic" is the one that presents greater development of the aerial part and yet less root growth. At the level of material and methods, the nutritional composition of all organic fertilizers, and of the "starter" fertilizer included in the substrates, should be included, as well as when they have been applied (it is not entirely clear with certain contradictory paragraphs). Additionally, it should include how much nutrient each plant receives, not just the dose applied. At the level of results, there is a lot of information for which comparative statistics of the different treatments have not been carried out, making it difficult to obtain conclusions. The quality of tables, photos and figures should be improved, including a detailed legend. The discussion includes questionable reflections from my point of view, as they have not been clearly demonstrated with the data presented. In short, the conclusions indicated do not reflect the reality of the data, and a more in-depth study should be carried out, and desirably study the crop throughout its cycle. I think that the focus of the manuscript should be based on the differences in the nature and composition of each fertilizer, and this is what should be emphasized. More details are included in the attached document.

Experimental design

As I have indicated, the experimental design should have been completed, carrying out the study in the complete crop cycle and determining its yield.

Validity of the findings

The indicated conclusions are questionable as I have already indicated.

Additional comments

See the attached document.

Annotated reviews are not available for download in order to protect the identity of reviewers who chose to remain anonymous.

---

## Round 0.2 · Minor Revisions

Dear Authors, thanks for the improvement of the manuscript that needs now only minor revision

Reviewer 1 ·

Basic reporting

no comment

Experimental design

no comment

Validity of the findings

no comment

Additional comments

I believe the authors address my concerns and i have no additional comments

Reviewer 2 ·

Basic reporting

First of all, I would like to thank the authors for their detailed responses to my questions. I have noted that most of the suggestions have been included in the manuscript. Despite the lack of yield results, which I suggest you check in future research, I feel that the manuscript is sufficiently detailed for publication.

Experimental design

Sufficiently detailed.

Validity of the findings

I suggest that the conclusions state that future studies will determine the validity of the fertilisers and doses applied based on crop yield.

Additional comments

No additional comments.

---

## Round 0.3 · accepted · Accept

I confirm the authors have addressed all of the reviewers' comments and the manuscript is now ready to be accepted.